# The Role of Neutrophils in Oncolytic Orf Virus-Mediated Cancer Immunotherapy

**DOI:** 10.3390/cells11182858

**Published:** 2022-09-14

**Authors:** Jessica A. Minott, Jacob P. van Vloten, Lily Chan, Yeganeh Mehrani, Byram W. Bridle, Khalil Karimi

**Affiliations:** Department of Pathobiology, University of Guelph, Guelph, ON N1G 2W1, Canada

**Keywords:** oncolytic virotherapy, cancer, Orf virus, neutrophils, antitumor immunity

## Abstract

Neutrophils are innate leukocytes with diverse effector functions that allow them to respond to pathogens rapidly. Accumulating evidence has highlighted these cells’ complex roles in the host’s response to viral infections and tumor progression. Oncolytic virotherapy is emerging as a promising treatment modality in the armamentarium of cancer therapeutics. Oncolytic viruses preferentially kill cancer cells and stimulate tumor-associated inflammation, resulting in tumor regression. Assessing the activity of individual effector cell subsets following oncolytic virotherapy is important in identifying their contribution to antitumor immunity. In this study, we investigated the role of neutrophils in oncolytic Orf-virus-mediated immunotherapy in a murine model of pulmonary melanoma metastases. The systemic administration of the Orf virus stimulated a dramatic increase in the number of leukocytes in circulation and within the tumor microenvironment, most of which were neutrophils. Analysis of tumor-burdened lungs shortly after therapy revealed significant numbers of phenotypically immature neutrophils, with the enhanced expression of molecules affiliated with activation, migration, and cytotoxicity. Neutrophils stimulated by Orf virus therapy were directly tumoricidal through tumor necrosis factor-α-mediated effects and were required for optimal antitumor efficacy following Orf virus therapy. Taken together, these data reveal neutrophils as a crucial innate effector to consider when investigating oncolytic virotherapy.

## 1. Introduction

Neutrophils are a class of short-lived, specialized innate leukocytes that act as rapid responders to invading pathogens [1,2]. They represent the most abundant granulocytic myelocyte-derived cell population in the blood and possess a diverse array of effector functions, such as the secretion of cytotoxic granules, phagocytosis, antigen transport and presentation, and the secretion of cytokines and chemokines that allow for the recruitment and activation of other effector leukocytes [3]. As a result, neutrophil responses have been implicated in several physiological and pathological conditions, including sterile injury, autoimmunity, viral infections, and cancers [3].

While antiviral immunity has historically been affiliated with adaptive immune responses, much evidence now exists to highlight the critical roles that the effector cells of the innate system play in the host response to viruses. The antiviral capabilities of neutrophils are broad and include the phagocytosis of virus-infected cells, the release of neutrophil extracellular traps, the secretion of reactive oxygen species, cytotoxic granules, and inflammatory molecules that serve to limit viral infection and spread, and mediate the initiation of adaptive antiviral processes [4]. Neutrophils have also emerged as important players in tumorigenesis, with conflicting evidence highlighting the capacity of these cells to exhibit both pro- and antitumoral properties depending on their environmental stimuli and phenotypic polarization [5,6]. The heterogeneous and plastic nature of these cells endows them with the ability to exhibit both beneficial and detrimental tumor responses, such as direct tumor cell cytotoxicity and the orchestration of adaptive antitumor immunity, or the promotion of tumor cell proliferation, angiogenesis, and metastasis [2,7,8]. Considering the effects neutrophils can have in the context of both viral infection and tumorigenesis, it is highly relevant to investigate their response to oncolytic virotherapy.

Of the many immunotherapies that exist for the treatment of cancers, oncolytic viruses (OVs) have emerged as strong contenders and promising alternatives for overcoming the limitations commonly associated with traditional anticancer therapies. OVs comprise a diverse group of anticancer viruses that can initiate potent antitumor responses through the preferential targeting and lysis of tumor cells in an immunogenic fashion [9]. The advantages of these viruses include their ability to kill tumor cells directly, stimulate intratumoral inflammation, and trigger both the innate and adaptive tumor-targeting immune responses [10,11,12,13]. One of the OVs that has been explored includes the replication-competent poxvirus Orf virus (OrfV), which can potently kill human and murine cancer cells in vitro and has demonstrated efficacy in vivo, as observed through a reduction in tumor burden, extended survival, and enhanced antitumor immune responses [14,15]. The efficacy of OrfV as an anticancer therapy has been primarily attributed to the potent induction of tumoricidal natural killer (NK) and T-cell responses [14,16,17], but the contribution of neutrophils to the overall antitumor response elicited following OrfV therapy remained to be elucidated.

While the varied functions of neutrophils in tumor pathogenesis have been established, research on the phenotype and activity of neutrophils following OV-based immunotherapies in tumor-burdened environments remains sparse. Given the unique plasticity of these cells and their ability to exert both pro- and antitumor effects as well as antiviral effects, and the complex pattern of interactions between neutrophils and other leukocytes, understanding how neutrophils function in response to OVs in a tumor-burdened setting could provide important insights into the therapeutic outcomes of OV therapies. In this study, we aimed to investigate the impact of oncolytic OrfV therapy on neutrophil phenotype and function in a pre-clinical lung metastasis model of melanoma. We found compelling evidence of OrfV-mediated antitumor neutrophil activity. We demonstrated that OrfV potently activated neutrophils, primarily immature in phenotype, and promoted their infiltration into the TME. Neutrophils were implicated in enhancing viral amplification, reducing tumor burden, and extending survival following OrfV therapy. These experiments uncovered an unexpected role of OrfV-stimulated immature neutrophils in antitumor cytotoxicity, which was mediated through the secretion of tumor necrosis factor (TNF)-α. These results provide a foundational understanding of the previously underappreciated role that neutrophils play in oncolytic virotherapy, particularly as mediated by OrfV.

## 2. Materials and Methods

### 2.1. Ethics and Biohazard Certification

Animal experimentation was approved under animal utilization protocol #4662 by the Animal Care Committee at the University of Guelph, Guelph, Ontario, Canada. All the experiments were conducted following the guidelines from the Canadian Council on Animal Care. Virus-related research was conducted in certified containment level-2 facilities under biohazard permit #A-367-04-19-05 issued by the University of Guelph’s Biosafety Committee.

### 2.2. Mice

Eight-week-old female C57BL/6 mice (Charles River Laboratories, Wilmington, MA, USA, strain code #027) were housed four in a cage in the animal isolation unit at the University of Guelph under specific pathogen-free and environmentally controlled conditions. Food and water were provided ad libitum. The mice were acclimatized to the facility for two weeks prior to experimentation. All the experiments were performed in triplicate, with *n =* 8 mice per group.

### 2.3. Virus

The NZ2 strain of Orf virus (OrfV) was kindly provided by Dr. Andrew Mercer (University of Otago, New Zealand). OrfV was produced and titrated by a 50% tissue culture infectious dose (TCID_50_) assay on sheep skin fibroblasts cultured in Dulbecco’s High-Glucose Modified Eagle’s Medium (DMEM; HyClone, catalog (Cat)#SH3002201) containing 10% heat-inactivated (HI) fetal bovine serum (FBS), as previously described [18].

### 2.4. In Vivo Tumor Model and Oncolytic Virotherapy

To establish a murine melanoma lung model, B16F10 cells (ATCC, Manassas, VA, USA, Cat#CRL-6475) were confirmed to be mycoplasma-free (MycoAlert PLUS detection kit, Lonza, Basel, Switzerland, Cat#LT07-705) and grown in DMEM supplemented with a 10% HI bovine calf serum (VWR International, Mississauga, ON, Canada, Cat#2100-500) in a humidified incubator at 5% CO_2_ and 37 °C. Then, 3 × 10^5^ cells were injected into the tail vein of the mice. The mice were then injected intravenously with phosphate-buffered saline (PBS; control group) or 5 × 10^7^ plaque-forming units (PFUs) of OrfV in 100 μL of PBS (treatment group), four days following the tumor challenge. For the survival experiments, the mice were monitored until the signs of respiratory distress and illness were observed, following which they were euthanized. For the visualization of lung tumor burdens, the mice were euthanized 14 days post-tumor challenge, and their lungs were harvested. The number of superficial lung metastases was determined by the separation of lung lobes and enumeration under a dissection microscope (Leica, Richmond Hill, Ontario, Canada). 

### 2.5. Tissue Processing

The mice were euthanized 24 h following the administration of OrfV or PBS, and the blood, lungs, spleens, peritoneal lavage fluid, and livers were collected for flow cytometric analyses. The retro-orbital blood and peritoneal lavage fluid samples were collected into heparinized microtubes to prevent clotting and then were kept on ice during transport and processing. Volumes were recorded to facilitate normalizing the flow cytometry data on a per microliter basis, and erythrocytes were lysed using an ammonium–chloride–potassium (ACK) lysing buffer. Splenocytes were collected by pressing the spleens between the frosted ends of two glass microscope slides. Erythrocytes were lysed using ACK lysing buffer, and the total splenocytes were counted. Leukocytes were isolated from the lungs and livers through digestion with 1 mg/mL of collagenase type II (Sigma-Aldrich, St. Louis, MO, USA, Cat#234155-100MG) and 20 units of DNase I (Sigma-Aldrich, Cat#11284932001) for 30 min at 37 °C. The lungs and livers were then mechanically digested in gentleMACS C tubes (Miltenyi Biotec, Bergisch Gladbach, Germany, Cat#130-096-334) using a gentleMACS tissue dissociator (Miltenyi Biotec), following the dissociation protocols that were pre-programmed by the manufacturer. The samples were then filtered through a strainer with a 100 µM pore size (Thermo Fisher Scientific, Mississauga, ON, Canada, Cat#22363549) and plated for staining. When cytokine concentrations were analyzed, the samples were plated in Roswell Park Memorial Institute 1640 media (HyClone, South Loga, UT, USA, Cat#SH3002701) containing 10% HI FBS and 0.01% beta-mercaptoethanol and incubated for one hour before the addition of brefeldin A (eBiosciences, San Diego, CA, USA, Cat#00-4506-51). The samples were then incubated for an additional four hours before staining.

### 2.6. Analysis via Flow Cytometry

Neutrophils were quantified among the bulk leukocytes via the flow cytometric analysis of the blood, lungs, spleens, peritoneal lavage fluid, bone marrow, and livers 24 h following the administration of OrfV or PBS in B16F10 lung tumor-bearing C57BL/6 mice. The isolated cells had Fc receptors blocked (anti-CD16/32, BioLegend, San Diego, CA, USA, Cat#101320) for 15 min at 4 °C. For leukocyte analysis, the samples were divided and stained with FITC-CD45.2 (BioLegend, Cat#109806) for 20 min at 4 °C in the dark. Viability was assessed using a Zombie Aqua viability dye kit (BioLegend, Cat#423102), following the manufacturer’s protocol. Neutrophils were isolated on the basis of the surface expression of FITC-CD45.2 (BioLegend, Cat#109806), PE-Ly6G (BD Biosciences, Cat#551461), APC-Cy7-Ly6C (BD Biosciences, Cat#560596) and BV421-CD11b (eBioscience, Cat#48-0112-82). Further neutrophil phenotypic analysis was performed by staining across several panels for the following surface markers for 20 min at 4 °C in the dark: FITC-CD69 (BD Biosciences, San Jose, CA, USA, Cat#557392), PerCP5.5-CXCR2 (BioLegend, Cat#149307), PE-Cy7-CD101 (BioLegend, Cat#331014), APC-CXCR4 (BioLegend, Cat#146507), and APC-Cy7-MHC class II (eBioscience, Cat#17-5321-82). The cytokine secretion from neutrophils was examined following a four-hour treatment with GolgiPlug (BioLegend, Cat#420601). The cells were then treated with a fixation buffer (BioLegend, Cat#420801) and a permeabilization buffer (BioLegend, Cat#421002) and stained across several panels for intracellular PE-TNF-α (eBioscience, Cat#12-7321-82), PerCP5.5-IL-10 (BioLegend, Cat#505028), APC-IL-17a (eBioscience, Cat#17-7177-81), PE-IFN-γ (BioLegend, Cat#505808), APC-IL-12p40 (BioLegend, Cat#505206), FITC-Granzyme B (BioLegend, Cat# 396403), and APC-IL-6 (BD Biosciences, Cat#561367) for 20 min at 4°C in the dark. The cells were washed and suspended in 200 μL of a fluorescence-activated cell scanning buffer (PBS + 0.5% bovine serum albumin; Thermo Fisher Scientific, Mississauga, ON, Canada, Cat#BP1600100) for flow cytometry analysis. The samples were run on a FACS Canto II flow cytometer (BD Biosciences, ON, Canada) and analyzed using FlowJo Software version 10 (FlowJo LLC, Ashland, Catlettsburg, OR, USA).

### 2.7. Antibody-Mediated Depletion Studies

Briefly, 3 × 10^5^ B16F10 cells were injected into the tail vein of the mice. To deplete the neutrophils, one day and three days post-tumor challenge, a Ly6G-specific depleting antibody (Cedarlane, Burlington, ON, Canada, Cat#BP0075-1-50MG-A) was administered intraperitoneally at a concentration of 200 μg/200 μL. Four days post-tumor challenge, the mice were intravenously injected with PBS or 5 × 10^7^ pfu of OrfV in 100 μL of PBS. The mice were injected with the Ly6G-specific depletion antibody every two days thereafter to maintain the depletions. The depletions were confirmed by the flow cytometric analysis of the blood-derived leukocytes prior to administering the oncolytic virotherapy.

### 2.8. Virus Growth Curve

To examine the production of infectious OrfV following therapy in vivo, lung tumor-bearing mice were treated with 5 × 10^7^ PFU of OrfV. At 72 h post-virus delivery, the tumor-burdened lungs were homogenized using a Precellys 24 Automatic Homogenizer (Bertin Technologies, Rockville, ML, USA). The lungs were collected in 750 µL of DMEM supplemented with 10% HI FBS and transferred to lysing matrix tubes containing ¼-inch ceramic matrix M beads (MP Biomedicals, Solon, Ohio, USA; Cat#6910-100). The tissues were homogenized with two cycles at 5000 rpm at 4 °C. The cell lysates were subjected to three freeze–thaw cycles to disrupt the cell membranes and release intracellular virions. The clarified virus-containing supernatant was collected after 10 min of centrifugation at 10,000× *g* at 4 °C. The virus-containing supernatant was titrated by TCID_50_ on sheep skin fibroblasts, and the resulting TCID_50_ values were converted to PFUs by multiplying by 0.69, as previously described [19,20].

### 2.9. Neutrophil Cytotoxicity Assays

Neutrophils were isolated from the bone marrow of flushed femurs from tumor-free OrfV- or PBS-treated C57BL/6 mice two hours post-infection. The isolated cells were pooled, stained with anti-Ly6G-PE, the viability dye 7-aminoactinomycin D (Cat#420404), and anti-CD11b-BV421, and sorted for the neutrophils based on positive expression using a FACS Aria flow sorter. B16F10 cells were harvested and labeled with carboxyfluorescein diacetate succinimidyl ester (ThermoFisher Scientific, Cat#C34554) and plated at 3 × 10^5^ cells per well in a 96-well plate. The sorted neutrophils from the OrfV- or PBS-treated mice were co-incubated with target B16F10 cells at increasing target-to-effector ratios. The cells were incubated for 16 h prior to staining and the assessment of tumor cell viability via flow cytometry. To investigate the influence of TNF-α on neutrophil-mediated tumor cell death, the sorted neutrophils were incubated with a TNF-α-specific depletion antibody (BioLegend, Cat#506331) at 0.2 mg/mL per well prior to incubation with the target tumor cells.

### 2.10. Statistical Analyses

GraphPad Prism version 9 for Mac (GraphPad Software, San Diego, CA, USA) was used for all graphing and statistical analyses. The survival curves were plotted using the Kaplan–Meier method, and the differences between groups were queried using the log-rank Mantel–Cox test. The immune response data, which involved one variable, were assessed by one-way analysis of variance (ANOVA) with Tukey’s multiple-comparison test, or by Student’s two-tailed *t*-test. The data that involved two variables were assessed by two-way ANOVA with Tukey’s multiple-comparison test. All the reported *p*-values were two-sided and were considered significant at a *p*-value ≤ 0.05. The graphs show means and standard errors.

## 3. Results

### 3.1. Intravenous Delivery of OrfV Recruits Leukocytes to the Tumor Microenvironment and Activates a Systemic Neutrophil Response in Tumor-Bearing Mice

The immunotherapeutic potential of OrfV was tested using an immunocompetent B16F10 melanoma model of pulmonary metastases. The mice bearing four-day-old pulmonary metastases were treated intravenously with a single dose of PBS or 5 × 10^7^ PFU of OrfV. The blood, lungs, peritoneal lavage fluid, spleens, and livers were acquired 24 h later for the analysis of leukocytes via flow cytometry. Treatment with OrfV dramatically increased the number of leukocytes in tumor-bearing lungs (Figure 1a). Additionally, treatment with OrfV increased the number of leukocytes found in the blood, intraperitoneal cavity, spleens, and livers of tumor-bearing mice, compared with the PBS-treated controls, with neutrophils dominating a large proportion of the leukocytes seen across all the tissues analyzed (Figure 1b), as well as in circulation (Figure 1c). Further analysis of tumor-bearing lungs showcased an increase in the number of neutrophils 24 h following OrfV therapy (Figure 1d). These data indicated that treatment with OrfV could stimulate systemic and TME-localized leukocyte and neutrophil responses.

### 3.2. Phenotypically Immature Neutrophils with Increased Expression of Migratory and Activation Molecules Respond to Infection with OrfV to Colonize the Tumor Microenvironment of the Lungs

Given the observation that treatment with OrfV enhanced the recruitment of neutrophils into circulation and the TME of the lungs, the phenotypes of these neutrophils were evaluated further. To this end, the blood and B16F10 tumor nodule-bearing lungs were acquired from mice 24 h post-treatment with OrfV. Again, the numbers of neutrophils in circulation and the lungs dramatically increased following the administration of OrfV. Most of these neutrophils in the OrfV-treated mice were CD101^−^, which was suggestive of an immature phenotype (Figure 2a). Further phenotypic analysis of these immature neutrophils in circulation revealed an increase in the expression of the lung-homing chemokine receptor CXCR2 (Figure 2b) and the activation marker CD69 (Figure 2c). The proportion of the immature neutrophils in circulation that expressed major histocompatibility complex II (MHC II), which is involved in antigen presentation, remained relatively unchanged between the OrfV-treated mice and PBS-treated controls (Figure 2d). These observations indicated that OrfV induced the rapid activation and emigration of phenotypically immature neutrophils into circulation for localization within the TME of the lungs while maintaining innate antigen presentation capabilities.

### 3.3. Neutrophils Activated by OrfV Exhibited Enhanced TNF-ɑ-Mediated Cytotoxicity against Target Tumor Cells

It is well-documented that neutrophils possess an arsenal of effector functions that allow these cells to recognize, sequester, and kill pathogens directly. Thus, we performed a series of ex vivo cytotoxicity studies to assess whether OrfV-stimulated neutrophils could mediate tumor cell death. Tumor-free mice were treated with 5 × 10^7^ PFU of OrfV or PBS intraperitoneally. At 24 h post-infection, the neutrophils isolated from the OrfV-treated mice exhibited enhanced killing of target B16F10 cells compared with those isolated from the PBS-treated mice (Figure 3a). This result highlighted the ability of OrfV to enhance cytotoxic antitumor neutrophil responses.

To further elucidate the mechanism behind OrfV-mediated neutrophil antitumor cytotoxicity, the lung-derived cells from B16F10 metastasis-bearing mice were analyzed via flow cytometry at 24 h post-treatment with OrfV to quantify the neutrophil-derived cytokines. Importantly, these data were collected after a four-hour blockade of the Golgi apparatus-mediated export of proteins in the absence of any ex vivo stimulation. Following OrfV therapy, CD101^−^ neutrophils in the TME of the lungs exhibited the enhanced production of TNF-α, which was not observed in CD101^+^ neutrophils following OrfV treatment or in either of the neutrophil subsets from the PBS-treated control mice (Figure 3b). Given this observation, we hypothesized that the ability of OrfV-stimulated neutrophils to mediate target tumor cell death was a direct result of enhanced TNF-α secretion. To investigate this, tumor-free mice were treated with 5 × 10^7^ PFU of OrfV or PBS intraperitoneally. At 24 h post-infection, the neutrophils isolated from the OrfV- or PBS-treated mice were incubated with TNF-α-neutralizing antibodies prior to co-culture with the target tumor cells. The cytotoxicity of OrfV-stimulated neutrophils against B16F10 tumor cells was significantly reduced in the presence of anti-TNF-α, an effect that was not observed in the neutrophils isolated from the PBS-treated controls (Figure 3c). These data underscore the ability of OrfV to enhance the activation of cytotoxic neutrophils capable of directly killing tumor cells through TNF-α-mediated mechanisms.

### 3.4. Depletion of Neutrophils Implicated them in OrfV-Mediated Antitumor Efficacy

To examine the contribution of neutrophils to OrfV-mediated antitumor efficacy, they were depleted in vivo, and their impact on survival, tumor burden, and viral replication was measured. C57BL/6 female mice were challenged with B16F10 melanoma cells intravenously. Metastases-bearing mice were treated with a Ly6G-specific antibody to deplete neutrophils or isotype control and subsequently treated intravenously with PBS or 5 × 10^7^ PFU of OrfV. The mice that received OrfV exhibited a significant survival advantage over the PBS-treated control mice, which was abrogated in the OrfV-treated mice that were depleted of neutrophils (Figure 4a). The lungs were harvested 14 days post-tumor challenge to grossly visualize and quantify tumor burden. A drastic reduction in tumor burden was observed following the systemic delivery of OrfV compared with the PBS-treated controls (Figure 4b). The OrfV-treated mice that were depleted of neutrophils presented with relatively higher numbers of tumor lesions on the lungs compared with the OrfV-treated mice that maintained the expression of neutrophils but still exhibited reduced tumor burden, compared with the PBS-treated controls. The depletion of neutrophils did not significantly impact tumor burden in the PBS-treated control mice. The mechanisms by which OrfV was able to mediate a reduction in tumor burden may be attributed to virus amplification within tumor beds, allowing for enhanced oncolysis and the recruitment of effector leukocytes. Therefore, we aimed to investigate the impact of neutrophils on virus amplification within metastasis-bearing lungs. The tumor-bearing mice had neutrophils depleted using anti-Ly6G, or they were treated with an isotype control. This was followed by the intravenous administration of PBS or 5 × 10^7^ PFU of OrfV. Seventy-two hours post-treatment, their lungs were harvested, and virus titers were determined. Neutrophils appeared to aid oncolytic OrfV replication in the TME, as demonstrated by the higher number of replication-competent virions detected in the lungs of the OrfV-treated mice with an intact neutrophil population compared with the OrfV-treated mice that had neutrophils depleted (Figure 4c). Cumulatively, these data suggested that neutrophils played a role in controlling tumor burden, at least in part by promoting the replication of OrfV. The total data demonstrated that neutrophils are a major driver of OrfV-mediated therapeutic efficacy in this pre-clinical model.

## 4. Discussion

This study is the first to comprehensively analyze the immunological effect of oncolytic OrfV on neutrophils in the context of a murine model of simulated pulmonary melanoma metastases. Oncolytic immunotherapy relies on an OV’s ability to target and infect tumor cells, induce tumor cell damage or death, and recruit leukocytes to the tumor site to induce antitumor immunity within the immunosuppressive microenvironment of the tumor stroma [10,11,12,13]. The analysis of neutrophils performed in the experiments demonstrated that oncolytic OrfV triggered a systemic inflammatory response from neutrophils, including their enhanced expression of trafficking and activation markers, and infiltration of the TME and cytotoxic effector functions. It is possible that the rapid infiltration of neutrophils to the site of metastases following treatment with OrfV was representative of an innate antiviral response to the administration of the virus, as has been previously documented [21,22,23]. However, the neutrophil response to the administration of OrfV did not appear to limit therapeutic efficacy but instead directly correlated with enhanced viral amplification, reduced tumor burden, and extended survival. The balance of neutrophil antiviral versus antitumor activity following OV therapy appears to vary significantly depending on the tumor and type of the OV under investigation [24,25]. In the various models of B-cell-derived cancers in the mice treated with oncolytic measles virus (MV), the depletion of neutrophils prior to treatment significantly abrogated the MV-mediated antitumor effect in a Raji–Burkitt lymphoma model but had no effect in a Nalm-6 acute lymphoblastic leukemia model. This appeared to be related to the antiviral response elicited at tumor sites following the administration of MV, resulting in the polarization of neutrophils toward an N1 phenotype, reflected through increased activation, prolonged survival, inflammatory cytokine release, and degranulation [25,26]. While a reduction in MV titer was seen, the antiviral neutrophil response indirectly aided the antitumor response elicited following therapy.

Similarly, the massive infiltration of activated neutrophils with enhanced cytokine secretion capabilities has been seen following the administration of an oncolytic herpes simplex virus (HSV) and a modified oncolytic MV in tumor-bearing mice [27,28]. The administration of an OV resulted in the initial influx of neutrophils to tumor sites, which correlated with tumor regression. An ex vivo investigation of tumor-infiltrating neutrophils following infection with HSV highlighted a significant increase in their tumor-killing capacity, compared with the neutrophils isolated from non-infected tumors [24]. The results of these studies showcased how the response of neutrophils to viral infections could indirectly induce antitumor effects in the TME.

In our study, the administration of OrfV induced the rapid recruitment of the leukocytes dominated by neutrophils to the site of metastases. Within 24 hours of infection, most recruited neutrophils exhibited an immature phenotype, as indicated by their lack of expression of CD101. However, the antiviral functional capacity of neutrophils at the different stages of maturation is poorly understood due to challenges in their isolation and identification and remains an open question relevant for future studies to explore. More recently, neutrophil maturity states have also been a subject of interest within the context of tumorigenesis. In many cases, immature neutrophils have been more commonly associated with suppressive, pro-tumor functions [29]. However, the differences in the tumor-promoting versus tumor-suppressing capacity of neutrophils in relation to their maturity state appear heavily influenced by tumor type, isolation, identification protocols, and the type of therapeutic intervention [29,30,31]. In the case of OrfV-mediated therapy, the observed pre-mature release of immature CD101^−^ neutrophils from the bone marrow into circulation for localization at tumor sites was most likely due to emergency granulopoiesis, triggered by the presence of infectious viral particles [32]. Once in the TME, OrfV infection increased the activation status and release of TNF-α by recruited CD101^−^ neutrophils, which we later observed to mediate the overall neutrophil cytotoxicity against the target tumor cells. It is important that future research further dissect the heterogeneity of neutrophil subsets and the role that the maturity status plays in the modulation of tumors, to form the basis for more targeted approaches that enhance the populations that drive tumor control while suppressing those that contribute toward tumor progression following OV therapy.

TNF-α has been implicated in both mediating potent antiviral effects and the cytotoxicity of tumor cells [33]. For example, the viral activation of Toll-like receptor (TLR) signaling can trigger neutrophils to exert enhanced antiviral activities, including an increased expression of IL-6, which has previously been shown to coincide with the enhanced production and release of TNF-α by neutrophils [34]. While in this study, we did not notice an increase in the secretion of any cytokines commonly affiliated with TLR-induced signaling by neutrophils, such as IL-6, IFN-γ, or IL-10, thus suggesting that the increase in TNF-α secretion by neutrophils seen following OrfV administration was TLR-induced, it would be interesting to investigate the induction of the TLR-signaling pathways activated by OrfV infection, and the subsequent impact this may have on neutrophil antiviral versus antitumor activity within the TME. TNF-α-secreting neutrophils have previously been shown to mediate the antitumor efficacy of an oncolytic poliovirus in the xenograft models of breast and prostate cancers [35]. In our study, TNF-α was implicated in mediating the OrfV-induced neutrophil cytotoxicity against tumor cells ex vivo. Exactly how significant neutrophil-derived TNF-α is in mediating the antitumor effects elicited by OrfV remains tenuous, and future in vivo research aimed at demonstrating a functional requirement for neutrophil-derived TNF-α in OV-mediated antitumor efficacy is warranted. On this basis, the immuno-modulation of the TME by the concomitant administration of TNF-α with OV therapy may augment antitumor neutrophil responses and be an additional treatment strategy worth considering. 

There is evidence that various OVs can directly infect and replicate within innate leukocytes such as neutrophils [22,28,35,36,37], which can not only potentiate OV activity but has also been shown to lead to prolonged neutrophil activation and the enhanced secretion of important antitumor cytokines, such as TNF-α, that have direct antitumor effects [28]. The exploitation of neutrophils for virus dissemination and amplification has been seen following infection with West Nile virus, human cytomegalovirus, Epstein–Barr virus, and influenza virus, whereby neutrophils were revealed to either systemically shuttle viral particles or productively support viral infection and replication [38,39,40,41]. The removal of neutrophils using a monoclonal antibody in our experiments showcased a strong correlation between the presence of neutrophils and OrfV-mediated antitumor efficacy, as observed through reduced viral amplification, increased tumor burden, and abrogated survival upon the depletion of neutrophils in the OrfV-treated mice. An important question remaining to be addressed is whether neutrophils serve as “off-targets” for OrfV-mediated infection and replication, which consequently allows for increased viral amplification and spread at tumor sites and the indirect enhancement of the antitumor efficacy elicited by OrfV. Future studies should investigate whether replication-defective OrfV induces similar neutrophil responses as replicating viruses to confirm this possibility. Additionally, investigating the OrfV viral burden in other tissues or serum will be informative to determine if the neutrophil-mediated enhancement of virus replication is lung-tissue-specific or systemic. Furthermore, it is important to consider the impact of the depletion of neutrophils on the development of adaptive antitumor responses following OV therapy. Neutrophils have been shown to regulate the recruitment and functions of natural killer (NK), dendritic, and T and B cells through multiple mechanisms, many of which are still being elucidated [42,43]. In a pre-clinical model of melanoma, the antitumor effects of oncolytic vaccinia virus strains were directly mediated by antitumor NK cells and tumor antigen-specific T-cell responses, which directly correlated with the increased numbers of intratumoral neutrophils [44]. Thus, future research should also aim to determine whether the removal of neutrophils leads to abrogated antitumor responses following OrfV-vectored therapy as a result of limiting the antitumor activities of other leukocytes.

The role of neutrophils in oncolytic immunotherapy is complex and multifaceted. While the direct cytotoxic effects of neutrophils may restrict intratumoral virus propagation, spread, and therapeutic efficacy, neutrophil-mediated inflammation and cytotoxicity may be required for antitumor immune responses. The complexity of the data herein provides numerous avenues for exploring the antitumor effector phenotype and activity of neutrophils following OV therapy in more detail.

## 5. Conclusions

The ways in which OVs interact with innate leukocytes during virotherapy are nuanced and reflect the ability of these cells to modulate their behavior and function in response to the stimuli they are exposed to. The data presented herein established that oncolytic OrfV therapy in a pre-clinical model of pulmonary melanoma metastases resulted in the rapid recruitment of leukocytes, and especially neutrophils, to the TME. The phenotypically immature neutrophils were activated by OrfV and trafficked via the bloodstream to colonize the TME of the lungs, whereby they secreted TNF-α and exhibited enhanced cytotoxic effects against tumor cells. Additionally, the presence of neutrophils correlated with the replicative and therapeutic ability of OrfV to extend survival and reduce lung tumor burden. The mechanisms by which neutrophils contribute to the OrfV antitumor efficacy are complex in nature, and expanding our knowledge of how neutrophils respond to oncolytic virotherapy is vital to understanding how they modulate the host immune systems to improve disease outcomes. While many of the current OV studies focus on investigating adaptive immune responses, our data suggest that neutrophils should be an important consideration for optimizing the efficacy of OV-based treatment regimens in the clinic. By exploiting the nuances of the innate immune response to OV administration, future approaches to virotherapy can combine the optimization of innate immune dynamics with other promising modalities such as prime–boost strategies and adoptive cell therapies, to maximize the strength and durability of the antitumor effect elicited by OVs.

## Figures and Tables

**Figure 1 cells-11-02858-f001:**
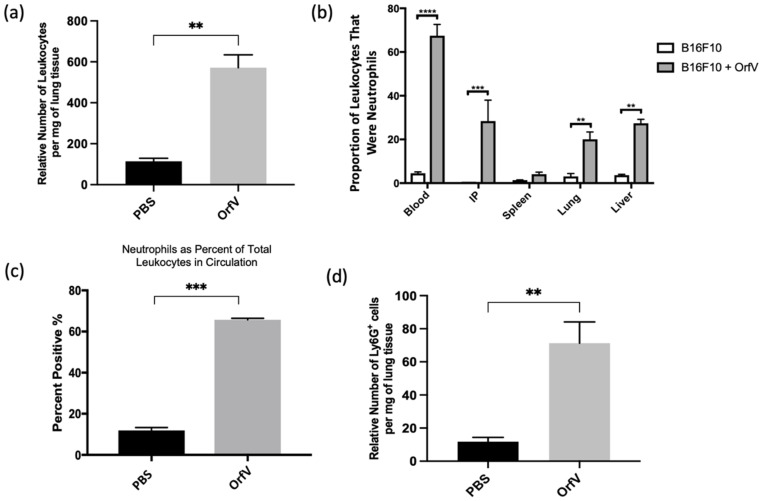
Intravenous delivery of OrfV recruited leukocytes to tumor-bearing lungs and activated systemic neutrophil responses. Seven- to nine-week-old female C57BL/6 mice (*n* = 8 per group) were injected intravenously with 3 × 10^5^ B16F10 melanoma cells to simulate pulmonary metastases. Four days post-tumor challenge, mice were injected intravenously with 5 × 10^7^ plaque-forming units of OrfV or phosphate-buffered saline (PBS) and euthanized 24 h later: (**a**) the number of leukocytes in the lungs was quantified via flow cytometry based on size exclusion and surface expression of the pan-leukocyte marker CD45; (**b**) the proportion of neutrophils, designated as CD45^+^ Ly6G^hi^ CD11b^hi^, relative to total live cells in the blood, intraperitoneal cavity, spleen, lungs, and liver was determined via flow cytometry; (**c**) the percentage of CD45^+^ leukocytes that were Ly6G^hi^ CD11b^hi^ neutrophils in circulation was quantified through flow cytometry; (**d**) the number of neutrophils in the lungs was quantified through flow cytometry based on having a CD45^+^ Ly6G^hi^ CD11b^hi^ phenotype. Statistical analysis was performed with two-tailed Student’s *t*-test for (**a**,**c**,**d**) and one-way analysis of variance in the case of (**b**). Statistical significance was designated as ** *p* ≤ 0.01, *** *p* ≤ 0.001, and **** *p* ≤ 0.0001.

**Figure 2 cells-11-02858-f002:**
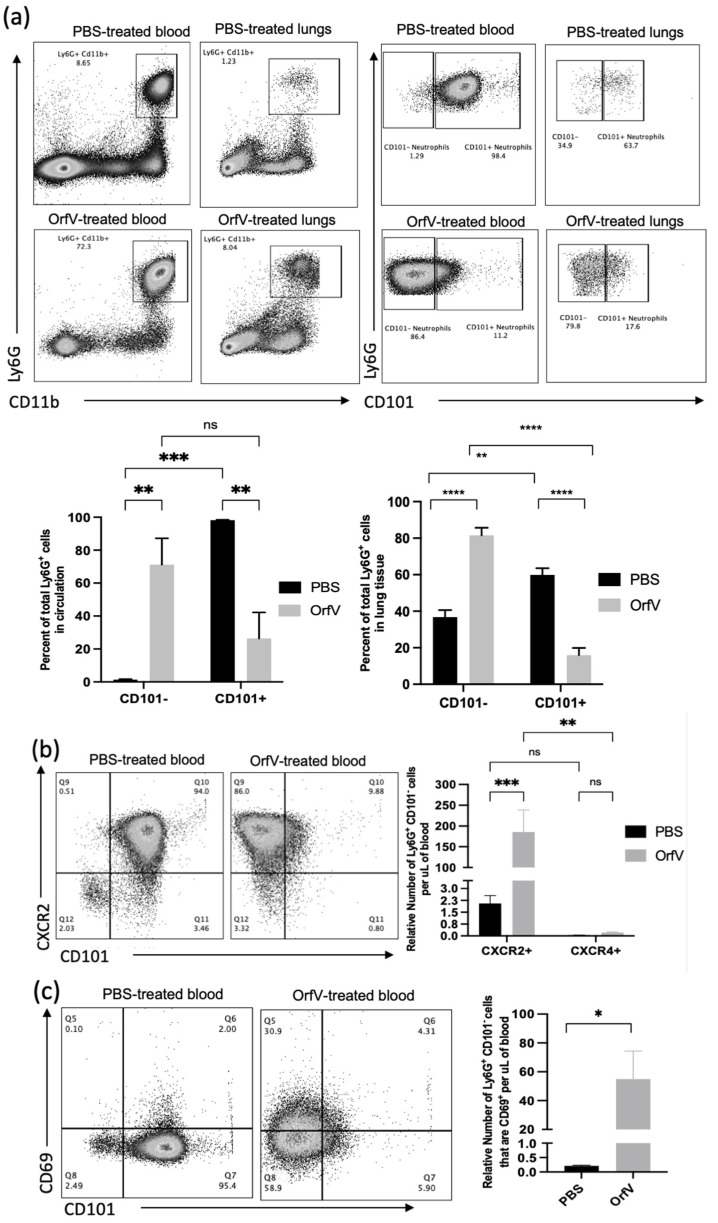
Immature neutrophils with increased surface expression of migratory and activation molecules entered circulation and colonized the TME of the lungs following OrfV therapy. Twenty-four hours after treatment with OrfV or PBS, blood, and lungs from pulmonary B16F10 metastasis-bearing seven- to nine-week-old female C57BL/6 mice were collected (*n* = 8 per group), and neutrophils were quantified with flow cytometry. Representative dot plots and graphs with standard errors are shown for (**a**) neutrophils (defined as CD45^+^ Ly6G^hi^ CD11b^hi^), and their corresponding maturity status (defined by expression of CD101); (**b**) phenotypically immature (CD101^−^) neutrophils expressing the lung-homing chemokine receptor CXCR2 in circulation; (**c**) activation status of circulating immature neutrophils (defined by expression of CD69 and CD101); (**d**) immature neutrophils expressing the antigen presentation molecule major histocompatibility complex class II (MHC II). Statistical analysis was performed using two-way analysis of variance in the case of (**a**,**b**) and by a two-tailed Student’s *t*-test for (**c**,**d**). Statistical significance was designated as * *p* ≤ 0.05, ** *p* ≤ 0.01, *** *p* ≤ 0.001, and **** *p* ≤ 0.0001.

**Figure 3 cells-11-02858-f003:**
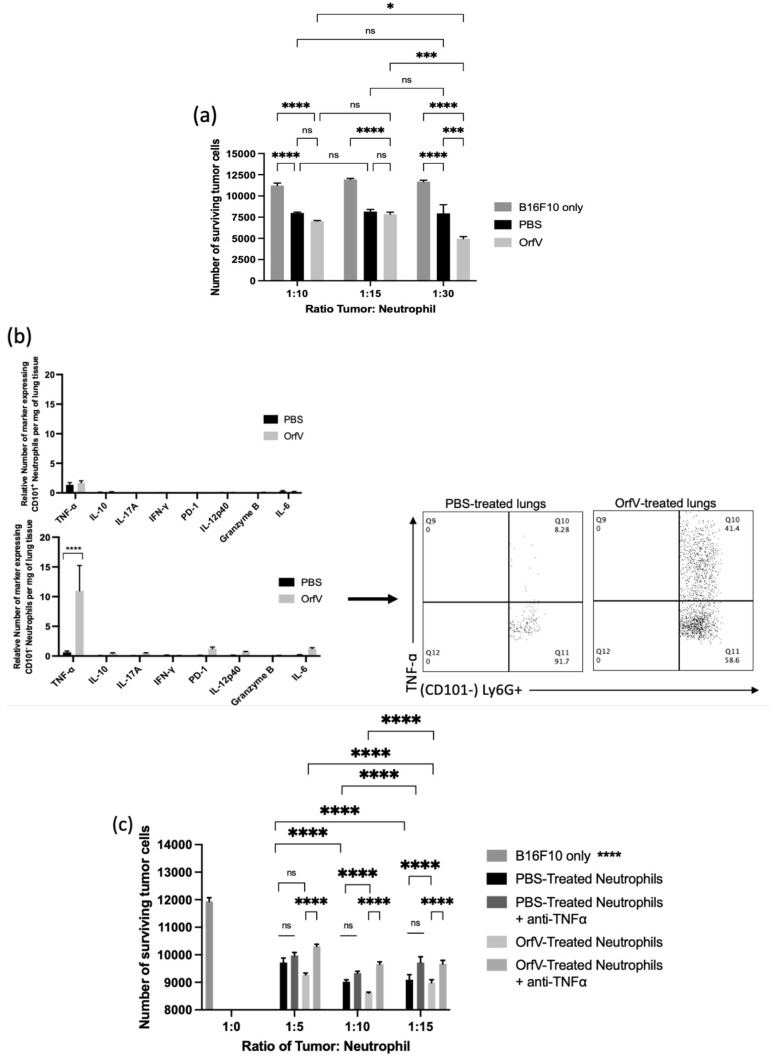
OrfV-stimulated neutrophils exhibited enhanced tumor necrosis factor (TNF)-ɑ-mediated cytotoxicity against tumor cells: (**a**) naïve seven- to nine-week-old female C57BL/6 mice (*n* = 8 per group) were injected intraperitoneally with phosphate-buffered saline (PBS) or 5 × 10^7^ PFU of OrfV. Two hours following treatment, bone marrow was flushed from femurs to isolate viable neutrophils (defined by a CD45^+^ Ly6G^hi^ CD11b^hi^ phenotype) by flow cytometric cell sorting. Neutrophils were co-cultured in vitro with B16F10 melanoma cells at increasing tumor-to-neutrophil ratios for 16 h. The viability of tumor cells was quantified via flow cytometry; (**b**) twenty-four hours after intravenous treatment with OrfV or PBS, B16F10 metastasis-bearing lungs were collected from the mice, and the cytokine response of immature and mature neutrophils (defined by CD101 expression) was quantified with flow cytometry after intracellular cytokine staining. TNF-α, tumor necrosis factor-alpha; IL-10, interleukin 10; IL-17A, interleukin 17A, IFNγ, interferon-gamma; PD-1, programmed cell death protein 1; IL-12p40, interleukin 12p40; IL-6, interleukin 6; (**c**) two hours after treating mice with OrfV, bone marrow-derived neutrophils (CD45^+^ Ly6G^hi^ CD11b^hi^) were flow-sorted. The isolated neutrophils were incubated with TNF-ɑ-neutralizing antibodies and co-cultured in vitro with B16F10 melanoma cells at increasing tumor-to-neutrophil ratios for 16 h. The cell viability of tumor cells was quantified with flow cytometry. Statistical analysis was performed using two-way analysis of variance. Statistical significance was designated as * *p* ≤ 0.05, *** *p* ≤ 0.001, and **** *p* ≤ 0.0001.

**Figure 4 cells-11-02858-f004:**
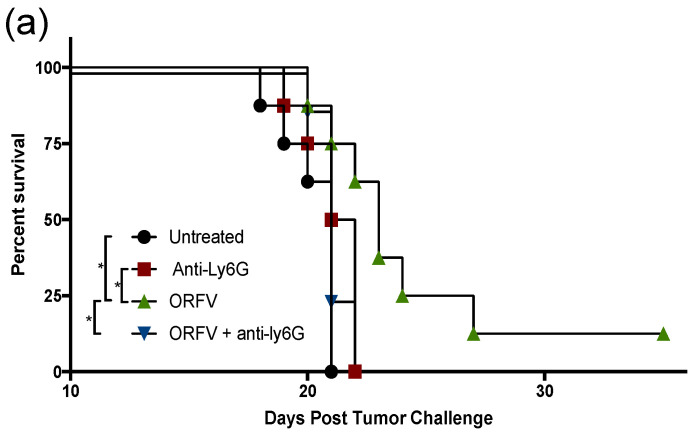
Depletion of neutrophils abrogated the extension of survival and influenced tumor burden and viral replication in the tumor microenvironment following Orf virus (OrfV)-mediated oncolytic virotherapy. Eight-week-old female C57BL/6 mice bearing B16F10 lung metastases (*n* = 8 per group) were treated with Ly6G-specific antibodies to deplete neutrophils. Controls did not have neutrophils depleted. Mice were then treated with PBS or 5 × 10^7^ PFU of OrfV via intravenous injections four days post-tumor challenge: (**a**) survival was assessed by a log-rank test; (**b**) fourteen days post-tumor challenge, lungs were excised, and the number of surface metastases was counted. Representative images of lungs from each group of mice are shown, and data are represented as a bar graph with standard errors; (**c**) twenty-four hours following administration of OrfV, lung tissues were excised, and replication-competent virions were enumerated using a tissue culture of 50% infective dose assay. The amount of input virus was subtracted from the results for all mice to show how much replication had occurred in vivo. Statistical analysis was performed using one-way analysis of variance. Statistical significance was designated as * *p* ≤ 0.05, ** *p* ≤ 0.01, *** *p* ≤ 0.001.

## Data Availability

Data are available upon reasonable request.

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
