# Peer review of "The Role of Neutrophils in Oncolytic Orf Virus-Mediated Cancer Immunotherapy"

_cells, 2022, doi:10.3390/cells11182858_

Round 1
Reviewer 1 Report
Neutrophils are playing very important role in tumorigenesis as their pro- and anti-tumoral properties are depending on the environment and on the phenotypic polarization. I would like to congratulate the group for their work; well designed, addressing relevant points according to the study proposal.
This paper focused on the impact of oncolytic OrfV therapy on neutrophil phenotype and function in a pre-clinical lung metastases model of melanoma. I find this study important as the results are sowing compelling evidence of OrfV-mediated antitumor neutrophil activity with specific role of OrfV-stimulated immature neutrophils in anti-tumor cytotoxicity.
Murine melanoma lung model, B16F10 cells and in vivo model is appropriated
Authors could Clarify :
1) how exactly the neutrophil depletion was performed
2) how the Neutrophils were isolated from lung, from the bone marrow and from pulmonary metastases.
3) It would be significant addition to the results to add some histological analyses particularly for the recruitment of leukocytes by neutrophils to the site of metastases because neutrophils have been shown to regulate the recruitment and functions of NK, dendritic, T-and B-cells. I presume that tissues are available or easy to collected them.
4) It would be interesting to discuss that an increasing number of studies have documented that TLR-induced cytokine expression by neutrophils can be positively/negatively influenced by immuno modulating factors such as IFNγ and IL-10, respectively and OrfV can activate Toll-like receptor signaling, and triggers the release of IL-6 that has been identified as an important mediator of the storm cytokine.
Author Response
"Please see the attachment."

Reviewer 2 Report
In this research article, Minott et al., evaluate the role of neutrophils during oncolytic virotherapy in B16F10 mouse model of lung metastases. The authors first show that upon infection of mice with Orf virus in B16F10 model, there is highly elevated systemic neutrophil response at 24-hour post-virus infection and these neutrophils appear to have an immature phenotype. The authors then perform in vitro tumor cell cytotoxic assay to show that Orf virus activated neutrophils reduce tumor cell survival and suggest a possible mechanism by which neutrophils perform tumor cell killing could be TNF-a dependent. Finally, authors perform in vivo neutrophil depletion to show that Orf virus mediated survival of metastasized mice depends on neutrophils. Depletion of neutrophils result in reduced viral titers in the tumor bearing lung tissue suggesting that the in vivo mechanism of by which neutrophils act in oncolytic virotherapy could be by supporting viral amplification.
This is a well-done study and evaluates an important question on the role of innate immune cell type such as neutrophil in oncolytic virotherapy. Since, less is known about neutrophils exact function, this study provides some crucial evidence in their possible function. The introduction is very well written and easy to follow. The discussion section touches on important points of contention and addresses some of the drawbacks that can be investigated in future studies. Overall, the experiments are scientifically sound, conclusions seem appropriate and the article provides interesting findings on the role of neutrophils in oncolytic virotherapy of Orf virus.
I have following comments for authors to consider:
Major points:
1) Authors show neutrophil data at 24 hours post Orf virus infection. Why did authors choose 24-hour time-point? Is 24 hour the peak of neutrophil response? It will be nice to perform a temporal response to determine the peak of neutrophil response. Authors mention that neutrophils dominate the response, however authors do not show data on other innate immune cells. Is there increase specific to neutrophils or do other innate cell such as monocytes also increase after Orf virus infection in B16F10 model?
2) In figure legends throughout the paper, the authors do not mention the number of times the experiments were performed and number of mice used in each experiment. This is important information to add in each figure legends.
3) In Figure 3, authors isolate neutrophils from bone marrow in their assay. However, whether the phenotype of neutrophils after Orf virus infection is comparable to blood is not shown. It will be important to show the surface expression of markers CD101, CD69, CXCR2 also on bone marrow neutrophils.
4) The role of neutrophils mechanistically seems confusing. In their in vitro experiments, the authors suggest that neutrophil derived TNF-a might play a role (Figure 3c). However, in vivo, in Figure 4c) the authors show neutrophils support viral replication. Do neutrophils have direct anti-tumor activity via TNF-a or act via enhancing viral replication or both is not clear. Have authors tried blocking TNF-a in their in vivo model to see if that affects metastatic lesions or survival of mice? This might not tell TNF-a from neutrophils specifically is playing a role, but might address if TNF-a is important or not?
5) In Figure 4c), the authors show reduced viral burden in the lung tissue upon neutrophil depletion. Showing one time-point can be sometimes misleading because there might shift in the kinetics of viral replication or dissemination. In that sense, it will be nice to show viral burden data also other timepoints such as 24-hr post infection and maybe a late timepoint. Including other tissue or serum viral burden will be informative to determine if neutrophil mediated enhancement of virus replication is lung tissue specific or systemic?
6) Although authors mention in their method section that they checked if neutrophils were depleted in the blood after anti-Ly6G antibody mediated depletion. It will be important to show the flow plot to determine the extent of depletion. Along with blood, it will be good to show data of neutrophils were successfully depleted in the lung tissue, as it is more relevant.
Minor points:
1) In Figure 1: Figure 1c and Figure 1d appear switched – what is depicted in figures and figure legend does not match.
2) In Figure 2b) authors mistakenly write CXCR4 on bar graph? This should be CXCR2?
3) Figure 4a) The survival data x-axis starts from 10 days. This should ideally be day 0?
Author Response
"Please see the attachment."
